# Urolithin and Reduced Urolithin Derivatives as Potent Inhibitors of Tyrosinase and Melanogenesis: Importance of the 4-Substituted Resorcinol Moiety

**DOI:** 10.3390/ijms22115616

**Published:** 2021-05-25

**Authors:** Sanggwon Lee, Heejeong Choi, Yujin Park, Hee Jin Jung, Sultan Ullah, Inkyu Choi, Dongwan Kang, Chaeun Park, Il Young Ryu, Yeongmu Jeong, YeJi Hwang, Sojeong Hong, Pusoon Chun, Hyung Ryong Moon

**Affiliations:** 1College of Pharmacy, Pusan National University, Busan 46241, Korea; dltkdrnjs54@naver.com (S.L.); heejung0319_@naver.com (H.C.); dbwls7523@naver.com (Y.P.); king2046@hanmail.net (H.J.J.); godhot785@naver.com (I.C.); 3607@naver.com (D.K.); myceaun@nate.com (C.P.); iy2355@naver.com (I.Y.R.); dassabn@gmail.com (Y.J.); yjw4238@gmail.com (Y.H.); wwjd0912@naver.com (S.H.); 2Department of Molecular Medicine, The Scripps Research Institute, Jupiter, FL 33458, USA; sultanwardag@gmail.com; 3College of Pharmacy and Inje Institute of Pharmaceutical Sciences and Research, Inje University, Gimhae 50834, Korea; pusoon@inje.ac.kr

**Keywords:** tyrosinase, urolithin, reduced urolithin, 4-substituted resorcinol, anti-melanogenesis, docking simulation

## Abstract

We previously reported (*E*)-β-phenyl-α,β-unsaturated carbonyl scaffold ((*E*)-PUSC) played an important role in showing high tyrosinase inhibitory activity and that derivatives with a 4-substituted resorcinol moiety as the β-phenyl group of the scaffold resulted in the greatest tyrosinase inhibitory activity. To examine whether the 4-substituted resorcinol moiety could impart tyrosinase inhibitory activity in the absence of the α,β-unsaturated carbonyl moiety of the (*E*)-PUSC scaffold, 10 urolithin derivatives were synthesized. To obtain more candidate samples, the lactone ring in synthesized urolithins was reduced to produce nine reduced urolithins. Compounds **1c** (IC_50_ = 18.09 ± 0.25 μM), 1h (IC_50_ = 4.14 ± 0.10 μM), and 2a (IC_50_ = 15.69 ± 0.40 μM) had greater mushroom tyrosinase-inhibitory activities than kojic acid (KA) (IC_50_ = 48.62 ± 3.38 μM). The SAR results suggest that the 4-substituted resorcinol motif makes an important contribution to tyrosinase inhibition. To investigate whether these compounds bind to human tyrosinase, a human tyrosinase homology model was developed. Docking simulations with mushroom and human tyrosinases showed that 1c, 1h, and 2a bind to the active site of both tyrosinases with higher binding affinities than KA. Pharmacophore analyses showed that two hydroxyl groups of the 4-substituted resorcinol entity act as hydrogen bond donors in both mushroom and human tyrosinases. Kinetic analyses indicated that these compounds were all competitive inhibitors. Compound 2a inhibited cellular tyrosinase activity and melanogenesis in α-MSH plus IBMX-stimulated B16F10 melanoma cells more strongly than KA. These results suggest that 2a is a promising candidate for the treatment of skin pigment disorders, and show the 4-substituted resorcinol entity importantly contributes to tyrosinase inhibition.

## 1. Introduction

Melanin plays an important role by protecting the skin from harmful ultraviolet rays. However, abnormal accumulation of melanin leads to skin diseases, such as Riehl melanosis, melasma, and senile lentigo [1]. Numerous approaches have been made to reduce abnormal melanin accumulation by suppressing tyrosinase expression [2], maturation [3,4], and catalytic activity [5] and enhancing its degradation [6]. Many natural and synthetic compounds, such as hydroquinone, monobenzyl hydroquinone, arbutin [7,8,9,10,11], kojic acid [12], salicylhydroxamic acid [13], azelaic acid [14], rucinol [15], phenylethyl resorcinol [16], thiazolyl resorcinols [17], and corticosteroids [13,14], have shown good anti-melanogenic effects in cell-based assays [18,19,20,21,22,23,24,25,26,27,28,29,30]. However, most of them have severe side effects in animal and humans models, such as inadequate potency, cancer, permanent depigmentation, and dermatitis [31]. Hydroquinone and kojic acid are prohibited in most countries due carcinogenic effects in thyroid [1], nephrotoxicity [32], genotoxicity [33], and melanocyte cytotoxicity [34]. Arbutin is a natural glycosylated hydroquinone that is extracted from the bearberry plant of the genus *Arctostaphylos*. Due to its fewer side effects, arbutin is more widely used as a whitening agent than hydroquinone, and its anti-melanogenic effect is known to be due to the inhibition of tyrosinase. Arbutin is hydrolyzed to D-glucose and hydroquinone by skin microflora (*Staphylococcus aureus* and *Staphylococcus epidermidis*) [35], enzymes like α-glycosidase, and by temperature [36]. In addition, skin-lightening agents used clinically, including arbutin and hydroquinone, do not have strong whitening effects. Therefore, there is considerable need for new tyrosinase inhibitors with high efficacies but less or no side effects.

Many customers are cautious of purchasing synthetic cosmetics because of concerns about side effects. Therefore, researchers have often sought to find new whitening agents from natural sources such as bacteria, fungi, and plants, investigations of which have identified bioactive compounds with anti-tyrosinase. Ellagic acid (EA) is generated from ellagitannins (hydrolysable tannins), such as sanguiin H-6 (from strawberry and raspberry), pedunculagin (from walnut), and vescalagin (from oak-aged wine). Urolithins are EA-derived metabolites produced naturally by human gut microbiota from ellagitannins in diet [37]. Urolithins are 6*H*-dibenzo[*b*,*d*]pyran-6-one derivatives, that is, 6*H*-benzo[*c*]chromen-6-ones, with different hydroxyl substitutions. Chemically their basic structures can be coumarin or isocoumarin like. Urolithins constitute a metabolic family (A, B, C, D, E, M-5, M-6, and M-7) and are produced by the hydrolysis and decarboxylation of one of the two lactone rings of EA and the sequential removal of hydroxyls from different positions on the two phenyl rings [38]. Urolithins have therapeutic effects such as anti-atherosclerotic [39], neuroprotective [40], heme peroxidases-inhibitory [41], anti-malarial [42], antioxidant [43,44], anti-inflammatory [43,44], and estrogenic modulatory effects [45]. Urolithins are patented for the prevention and treatment of neoplastic diseases [46] and also induce PC12 (rat pheochromocytoma cells) apoptosis [47]. Several researchers have modified the structures and synthesized derivatives of urolithins with different biological activities, such as cholinesterase inhibition [48,49], neuroprotective effects [50], anticancer effects in colon [51,52], breast [53], and prostatic cancers [54], as well as human topoisomerase II inhibition [55].

To catch up with the recent advancement in human tyrosinase inhibitors [56,57,58], over the past decade, we have identified many tyrosinase inhibitors and have accumulated much structure–activity relationship data [59,60,61,62,63,64,65,66,67]. Based on these data, we concluded the (*E*)-β-phenyl-α,β-unsaturated carbonyl scaffold plays an essential role in tyrosinase inhibitory activity and that when the β-phenyl of the scaffold is a 4-substituted resorcinol (2,4-dihydroxyphenyl), derivatives with such scaffolds exhibit potent tyrosinase inhibitory activities (Figure 1). We wondered whether 4-substituted resorcinol alone could impart tyrosinase inhibitory activity in the absence of the α,β-unsaturated carbonyl moiety of the scaffold. To explore this issue, we decided to investigate whether urolithin derivatives containing 4-substituted resorcinol inhibit tyrosinase (Figure 2). Therefore, we produced ten urolithin derivatives, including compounds 1c and 1f which contained 4-substituted resorcinol to study the influence of the 4-substituted resorcinol moiety on tyrosinase inhibition.

In addition, we considered reduction of the urolithin lactone ring could provide additional candidates for study. Therefore, *o*-phenylbenzyl alcohol derivatives including compounds 2a and 2e, which possess a 4-substituted resorcinol group, were prepared by reducing urolithin derivatives to produce compounds containing 4-substituted resorcinol, as depicted in Figure 2.

The synthesized urolithin derivatives and *o*-phenylbenzyl alcohol (reduced urolithin) derivatives were assayed for mushroom tyrosinase inhibitory activity, and compounds that caused potent inhibition were further studied for tyrosinase activity and melanin production in B16F10 murine melanoma cells and docking simulation. The results obtained supported the hypothesis that the presence of the 4-substituted resorcinol moiety importantly enhances tyrosinase inhibitory activity.

## 2. Results and Discussion

### 2.1. Chemistry

Urolithin derivatives have a 6*H*-benzo[*c*]chromen-6-one template. Hurtley synthesized 3-hydroxy-6*H*-benzo[*c*]chromen-6-one (urolithin B) by reacting 2-bromobenzoic acid with resorcinol under relatively mild conditions in 1929 [68]. However, this synthetic method was rarely utilized to produce benzo-coumarin until 2000. We synthesized urolithin derivatives 1a–1j using the Hurtley reaction [69], as previously described with minor modification [44]. Reactions between 2-bromobenzoic acid or 2-bromo-5-methoxybenzoic acid with resorcinol or phloroglucinol in the presence of CuSO_4_ in basic media gave urolithin derivatives 1a (urolithin B), 1c, 1g, or 1h in yields of 16–36% as depicted in Scheme 1. Other urolithin derivatives were obtained from these four urolithins. Methylation of the phenolic hydroxyl groups of 1a, 1c, 1g, and 1h using methyl iodide and K_2_CO_3_ produced urolithin derivatives 1b, 1d, 1i, and 1j, respectively and the demethylation of the phenolic methyl ether in 1g and 1h using AlCl_3_ furnished urolithin derivatives 1e (urolithin A), and 1f, respectively.

We envisioned that *o*-phenylbenzyl alcohol derivatives 2a–2i could be easily prepared from urolithin derivatives by reduction. The synthetic scheme used to produce *o*-phenylbenzyl alcohol derivatives is shown in Scheme 2 (Supplementary information are available in Appendix A). The target compounds 2a–2i were synthesized by reducing urolithin derivatives 1a–1e and 1g–1j using lithium aluminum hydride in yields of 47–76%. Unfortunately, the LiAlH_4_ reduced product of 1f could not be isolated due to its high polarity and purification difficulties.

### 2.2. Mushroom Tyrosinase-Inhibitory Activities of Urolithin Derivatives

Mushroom tyrosinase has been widely used as a target enzyme for screening potential tyrosinase inhibitors, and thus, the inhibitory activities of the synthesized urolithin and *o*-phenylbenzyl alcohol derivatives and kojic acid (the positive control; a well-known tyrosinase inhibitor) at a concentration of 50 μM were evaluated using mushroom tyrosinase. Table 1 summarizes the results for the synthesized urolithin derivatives. Although Su and co-workers [70] reported antimelanogenic effects through tyrosinase inhibition for urolithin A (1e) and urolitin B (1a) in B16F10 melanoma cells, urolithin A and urolitin B did not show inhibitory activity against mushroom tyrosinase in the present study (Table 1). Of the ten urolithin derivatives 1a–1j, two of the derivatives containing 4-substituted resorcinol, that is, 1c (68.93 ± 0.67% inhibition) and 1h (94.86 ± 0.16% inhibition) more potently inhibited tyrosinase than kojic acid (52.16 ± 2.05% inhibition), while 1f presented any inhibitory activity. The other urolithin derivatives, including urolithin A and urolithin B, exhibited no inhibitory activity. (compound 1f: 12.68 ± 6.09% inhibition). Urolithin derivatives commonly contain the 6*H*-benzo[*c*]chromen-6-one group, which is composed of three rings (Figure 1). Although urolithin B has a 3-hydroxyl group, it did not inhibit tyrosinase, whereas compound 1c, which has an additional hydroxyl group at the 1-position of urolitin B, greatly enhanced inhibition to 69%. Replacing the two hydroxyl groups of 1c and 1h with two methoxyl groups abrogated tyrosinase inhibitory activity (1c vs. 1d and 1h vs. 1j). The insertion of a hydroxyl group at the 8-position on compound 1c markedly reduced tyrosinase inhibition to 13% (1c vs. 1f). On the other hand, introduction of a methoxyl group at the same position surprisingly enhanced tyrosinase inhibition by 95% (1c vs. 1h). However, introduction of a methoxyl group at the 8-position of compound 1a resulted in no tyrosinase inhibitory activity. All urolithin derivatives without a hydroxyl group failed to inhibit tyrosinase (1b, 1d, 1i, and 1j). Taken together, the 1,3-dihydroxyl group in the C-ring of the 6*H*-benzo[*c*]chromen-6-one template, that is, the 4-substituted resorcinol motif played an essential role in the inhibition of tyrosinase and the 8-methoxyl group in the A-ring of the template played an auxiliary role, probably due to hydrophobic interactions with the amino acid residues of tyrosinase. As mentioned above, our several previous studies have demonstrated that (*E*)-β-phenyl-α,β-unsaturated carbonyl scaffold plays an important role in tyrosinase inhibitory activity and that derivatives with this scaffold generally potently inhibit tyrosinase when the β-phenyl of the scaffold is a 4-substituted resorcinol. Similar results were observed in the present study. Urolithin derivatives 1c and 1h, which possessed the 4-substituted resorcinol motif exhibited strong inhibitory activity. Compound 1f had a 4-sustituted resorcinol motif, but interestingly it only showed weak tyrosinase inhibition. Log P values were obtained by ChemDraw Ultra 12.0 and the log P values of all synthesized compounds were better than the standard kojic acid (Table 1).

### 2.3. Mushroom Tyrosinase-Inhibitory Activities of Reduced Urolithin Derivatives

To investigate whether the lactone ring (B-ring) of urolithin derivatives influences tyrosinase inhibition, the lactone ring was reduced to give *o*-phenylbenzyl alcohol derivatives 2a–2i. The lactone functional group was converted to hydroxymethyl and hydroxyl groups. Inhibitions of mushroom tyrosinase by the *o*-phenylbenzyl alcohol derivatives at 50 μM are summarized in Table 2. Of the nine *o*-phenylbenzyl alcohol derivatives 2a–2i, compound 2a showed strong inhibitory activity against mushroom tyrosinase (75.92% inhibition), which was greater than that of kojic acid (52.16 ± 2.05% inhibition). Like that shown by urolithin derivatives 1c and 1h, which contained the 4-substituted resorcinol motif, 2a, which also contained this motif most potently inhibited tyrosinase. Introduction of a 4′-hydroxyl group in 2a eliminated tyrosinase-inhibitory activity (2a vs. 2e). On the other hand, while introduction of an 8-methoxyl group into compound 1c increased inhibitory activity, insertion of a 4′-methoxyl group in 2a slightly decreased inhibitory activity (2a: 75.92% inhibition vs. 2f: 47.87% inhibition). The insertion of an additional 5-hydroxyl group into compound 2a produced a compound 2c with the 2-substituted phloroglucinol motif (Figure 1) and greatly diminished inhibitory activity to 19.78%. Energy minimization using Chem3D Pro 12.0 showed that unlike the urolithin derivatives, the A-ring and the C-ring of *o*-phenylbenzyl alcohol derivatives were no longer in the same plane (Figure 3). Compounds 2a (41.6°) and 2c (48.9°) had similar dihedral angle values of C(2′)-C(1′)-C(4)-C(3) (Figure 3). Although compound 2c had a conformation similar to 2a, their inhibitory activities were quite different (2a: 76% inhibition vs. 2c: 20% inhibition). This result implies that these different inhibitory activities were due to mismatched interaction between the 5-hydroxyl group and tyrosinase. The introduction of a 4′-methoxyl substituent in compound 2c increased inhibitory activity slightly (2c: 20% inhibition vs. 2h: 26% inhibition), as was observed for urolithin derivatives. All changes in the structures of resorcinol moieties in compounds 2a and 2f, such as the introduction of an additional hydroxyl or methoxyl or conversion of the hydroxyl to methoxyl reduced inhibitory activities (2a vs. 2b–2d, and 2f vs. 2g–2i). These results suggest that the 4-substituted resorcinol motif plays a critical role in the inhibition of tyrosinase activity. Log P values were obtained by ChemDraw Ultra 12.0 and the log P values of all synthesized compounds were better than the standard kojic acid (Table 2).

Taken together, our studies of urolithin and *o*-phenylbenzyl alcohol derivatives indicate that the 4-substituted resorcinol is a mandatory backbone for potent tyrosinase inhibitory activity and that the type, position, and number of substituents on the phenyl ring markedly influence abilities to inhibit tyrosinase.

### 2.4. Docking Studies Using Mushroom Tyrosinase Crystal Structure

The known tyrosinase inhibitors, kojic acid and tropolone, are representative competitive inhibitors that compete with substrates at the active site of tyrosinase. To examine whether compounds 1c, 1h, and 2a effectively bind to the active site of tyrosinase, docking simulation was conducted using AutoDock Vina 1.1.2 software developed by The Scripps Research Institute. Moreover, 3D-structures of 1c, 1h, and 2a were created by energy minimization of their 2D-ligand structures using Chem3D Pro 12.0 software (CambridgeSoft Corporation). The 3D structure of mushroom tyrosinase for docking simulation was obtained from *Agaricus bisporus* tyrosinase (PDB ID number: 2Y9X). Although the correlation between the binding affinities of the three ligands and degree of tyrosinase inhibition was not perfect, all ligands exhibited greater binding affinities (−7.6 ~ −6.9 kcal/mol) than kojic acid (−5.7 kcal/mol), a reference control (Figure 4e). LigandScout 4.2.1 software was utilized to examine interactions between the amino acid residues of tyrosinase and the functional moieties of ligands. As shown in Figure 4d, kojic acid interacts with amino acid residues of tyrosinase through two hydrogen bonds (His259 and His263) and one π-π stacking interaction (His263). Compound 1h creates four hydrogen bonds (His61, Asn260, His263, and His296) and two hydrophobic interactions with amino acid residues (Phe264 and Val283) of tyrosinase (Figure 4b), and compound 2a makes three hydrogen bonds (Asn260, Phe264, and Met280) and four hydrophobic interactions with amino acid residues (Val248, Phe264, and Val283) (Figure 4c). Compound 1c interacts hydrophobically with two amino acid residues (Val283 and Ala286) (Figure 4a). These results imply that like kojic acid, all three ligands bind to the active site of tyrosinase. However, LigandScout results did not explain why 1c binds more strongly to tyrosinase than 1h, 2a, and kojic acid. Therefore, two more docking simulation software packages, Dock 6 and AutoDock 4, were used to enhance the reliability of docking simulation results. The same tyrosinase species that were used for AutoDock Vina were utilized in these docking simulations. As indicated in Figure 5e, the binding affinities were −29.16, and −6.85 kcal/mol for 1c, −28.01, and −6.03 kcal/mol for 1h, and −30.15, and −6.68 kcal/mol for 2a, respectively, in Dock 6 and AutoDock 4, and all three had greater binding affinity than kojic acid (−27.29 kcal/mol in Dock 6 and −4.21 kcal/mol in AutoDock 4), as was observed in AutoDock Vina. Furthermore, these results were in good agreement with the results obtained during the mushroom tyrosinase inhibition experiment. According to results obtained using LigandScout, which is based on AutoDock 4 (Figure 5a–d), kojic acid creates one hydrogen bond with Met280 and one π-π stacking interaction with His263, which differed from that predicted by AutoDock Vina. The result of LigandScout based on AutoDock Vina indicated that kojic acid hydrogen bonds with His259 and His263. In addition, according to AutoDock 4 Met280 is involved in hydrogen bonding, whereas AutoDock Vina predicted His259 and His263 are involved in hydrogen bonding. In addition, the two programs predicted that two different hydroxyl groups of kojic acid are involved in hydrogen bonding (i.e., the branched hydroxyl group for AutoDock Vina vs. the ring hydroxyl group for AutoDock 4). In AutoDock 4, compound 1c makes two hydrogen bonds with His244 and Glu256 and two hydrophobic interactions with Val283 and Ala286, and compound 1h creates two hydrogen bonds with His244 and Glu256 and two hydrophobic interactions with Phe264 and Val283. Interestingly, although 1c and 1h hydrogen bond with the same amino acid residues, the hydroxyl groups of 1c and 1h that interact with these amino acids differ. Each hydroxyl group that interacts with these amino acids are opposite. While the two hydroxyl groups of the resorcinol moiety in 2a interact with amino acid residues through three hydrogen bonds in AutoDock Vina, AutoDock 4 showed 2a has four hydrophobic interactions with three amino acid residues (Phe264, Vla283, and Ala286) without hydrogen bonding. Taken together, the results of pharmacophore analyses obtained using LingandScout based on AutoDock Vina, and AutoDock 4 suggest two hydroxyl groups of the 4-substituted resorcinol participate in hydrogen bond formation at the active site of tyrosinase, and that the phenyl ring of the 4-substituted resorcinol participates in effective hydrophobic interactions. These results suggest that compounds containing the 4-substituted resorcinol moiety might be good candidates for tyrosinase inhibitors.

To further study the binding interactions and poses of compounds 1c, 1h and 2a, a docking study was performed using the Schrodinger Suite. Figure 6 depicts the 2D and 3D conformation of compounds 1c, 1h, 2a and kojic acid. The 4-substituted resorcinol of compound 1c forms two hydrogen bonds, as in the results obtained from AutoDock 4. However, interestingly, the resorcinol makes hydrogen bonds with different amino acids (Asn260 and Met280) from those (His244 and Glu256) involved in hydrogen bonding in AutoDock 4. Compound 1c is also involved in π-π stacking with Phe264. Whereas AutoDock Vina and AutoDock 6 docking results support that compound 1h is involved in multiple hydrogen bonding, Schrodinger Suite docking results support that compound 1h interacts with His85, Hie244, and His259 in a wide range of π-π stacking interactions without hydrogen bonding. It is noteworthy that two amino acids (His85 and His259), known to interact with two Cu(II) ions, are involved in the π-π stacking interaction with 1h. On the other hand, compound 2a produces only two π-π stacking interaction with His259 and His263, known to be involved in coordination with Cu(II) ions. Kojic acid which is used as a standard ligand creates one hydrogen bond with Gly281 and one π-π stacking interaction with His263. Based on the above interactions, the docking scores (Figure 7c) of compounds 1c, 1h, 2a, and kojic acid are −6.46, −5.25, −4.71, and −4.65 kcal/mol, respectively. The docking results obtained from Schrödinger Suite also show that compound 1c binds most strongly to mushroom tyrosinase. This result is consistent with that obtained from AutoDock Vina, and AutoDock 6. The docking results obtained from four docking simulation software consistently support that compounds 1c, 1h, and 2a bind to the active site of tyrosinase and inhibit tyrosinase activity more strongly than kojic acid and 4-substituted resorcinols result in high binding affinity for tyrosinase through multiple hydrogen bonds or multiple π-π stacking interactions.

### 2.5. Docking Studies Using Human Tyrosinase Homology Model

We were interested in whether our compounds could also be able to inhibit the human tyrosinase enzyme virtually. For this purpose, we developed a human tyrosinase homology model because the crystal structure for the human tyrosinase enzyme is not available to date. To build the human tyrosinase model, we used the Swiss Model online server and Schrödinger Suite Prime. The genome sequence of human tyrosinase (P146790) was imported from the UniProt Database [71] and the 3D homology model was built using the X-ray crystal structure of human tyrosinase related protein 1 (hTYRP1: 5M8Q) [72] as shown in Figure 7. The sequence identity of human tyrosinase and hTYRP1 was 45.81%. The homology model was further processed using the Schrödinger Suite and docked against compounds 1c, 1h, 2a, and kojic acid.

### 2.6. Docking Score and Binding Mode of Compounds **1c**, **1h**, **2a** and Kojic Acid at the Active Site of Human Tyrosinase Homology Model

A docking study was performed using the Schrödinger Maestro 2020-2 release to find the mode of inhibition of the compounds at the active site of the human tyrosinase homology model. The 2D and 3D structures of compounds 1c, 1h, 2a, and kojic acid are shown in Figure 8. Compound 1c produces one hydrogen bond with Ser380 and creates a π-π stacking interaction with His202. This resulted in a docking score of −4.82 kcal/mol, which means that 1c has higher binding affinity for mushroom tyrosinase (docking score: −6.46 kcal/mol) than human tyrosinase. The resorcinol moiety of compound 1h makes two hydrogen bonds with Ser375 and Ser380. In addition, A, B and C rings of compound 1h generates π-π stacking interaction with His202, Phe347, His363, and His367. These interactions resulted in a docking score of −5.42 kcal/mol, which is slightly better than the docking score of compound 1h (−5.25 kcal/mol) in mushroom tyrosinase. Compound 2a makes two hydrogen bonds with Glu203 and Ser380, and two π-π stacking interaction with Phe347 and His367. The docking score recorded for compound 2a is −5.07 kcal/mol, implying that 2a binds more tightly to human tyrosinase than mushroom tyrosinase (docking score: 4.71 kcal/mol). Kojic acid forms a metal coordination with a zinc ion (Zn7) through the branched hydroxyl group, a hydrogen bond with Ser375, and π-π stacking interaction with His367. The docking score of kojic acid is −4.15 kcal/mol, indicating that kojic acid has a weaker binding affinity than compounds 1c, 1h, and 2a in human tyrosinase homology model. Taken together, compounds 1c, 1h, and 2a behave in the human tyrosinase model in a manner similar to mushroom tyrosinase and occupy the same binding pockets at the active site of tyrosinase, such as kojic acid. These results suggest that compounds 1c, 1h, and 2a may be promising candidates for the treatment of hyperpigmentation disorders through human tyrosinase inhibition.

### 2.7. Binding Analysis of 1c, 1h, 2a and Kojic acid in a Human Tyrosinase Homology model and Mushroom Tyrosinase Enzyme

To further elaborate the interaction pattern of the compounds 1c, 1h, 2a, and kojic acid, the length of bonds and salts bridges was measured. As can be seen in Figure 9, the distance between the two copper ions and the 3-hydroxyl group of C-ring in compound 1c is 4.75 Å and 6.65 Å, which is relatively far compared to the corresponding distances (3.51 Å and 2.68 Å) in the ligand-protein complex between 2a and mushroom tyrosinase. In mushroom tyrosinase, the hydrogen bond length of 1c is 1.69 Å and 2.14 Å, whereas in human tyrosinase it is 1.93 Å. In compound 1h, the distance between the copper ions of mushroom tyrosinase and the 3-hydroxyl group of C-ring is 2.67 Å and 3.96 Å. And the distance between the zinc ions of human tyrosinase and 3-hydroxyl group of C-ring in compound 1h is 3.27 Å and 3.38 Å. The hydrogen bonds length of 1h in human tyrosinase is 1.91 Å and 2.11 Å. These hydrogen bonds may be related to the high binding capability of 1h towards human tyrosinase. In compound 2a, the distance between the copper ions of mushroom tyrosinase and the 1-hydroxyl group of C-ring is 3.51 Å and 2.68 Å, whereas the distance between the zinc ions of human tyrosinase and the corresponding hydroxyl group is 2.84 Å and 3.13 Å. These distances are too far to coordinate with the metal ions. The two hydrogen bond lengths in 2a are 2.07 Å and 2.13 Å, respectively. The higher binding affinity of 2a in human tyrosinase than in mushroom tyrosinase is probably due to the additional hydrogen bonding in human tyrosinase. In kojic acid, the distances between the Cu ions of mushroom tyrosinase and the vinylic hydroxyl group are 3.07 Å and 3.14 Å, these distances cannot be coordinating with the Cu ions. On the other hand, in the human tyrosinase homology model, kojic acid places its branched hydroxyl group near the Zn ions, instead of the vinylic hydroxyl group. The distances between the zinc ions of tyrosinase and the branched hydroxyl group are 2.13 Å and 2.44 Å, which means that kojic acid is located deeper in the active site in human tyrosinase than in mushroom tyrosinase. The branched hydroxyl group of kojic acid forms a metal coordination with a Zn(7) ion with a distance of 2.13 Å. For kojic acid, the hydrogen bond length of the branched hydroxyl group in mushroom tyrosinase is 2.13 Å and that of the vinylic hydroxyl group in human tyrosinase is 1.75 Å. From these interactions, the binding affinity of compound 1c decreases in human tyrosinase compared to mushroom tyrosinase, while the binding affinity of compounds 1h and 2a increases in human tyrosinase compared to mushroom tyrosinase. Overall, compounds 1c, 1h, and 2a exhibited superior binding affinity in both mushroom and human tyrosinase enzymes compared to kojic acid. It can be said that compounds 1c, 1h, and 2a are better able to inhibit the activity of tyrosinase in human tyrosinase than kojic acid. The results obtained from the 4 docking software imply that the 4-substituted resorcinol is a well-fitting structure for the active site of human and mushroom tyrosinase, due to its ability to hydrogen bond and/or pi-pi stacking interaction.

### 2.8. IC_50_ Evaluation of the Urolithin and Reduced Urolithin Derivatives

The inhibitory effects of 1c, 1h, 2a, and kojic acid on mushroom tyrosinase at several concentrations are provided in Table 3. All compounds inhibited tyrosinase activity in a concentration-dependent manner. Results indicated that 1h was the most potent mushroom tyrosinase inhibitor with an IC_50_ value of 4.14 ± 0.10 μM, which was 12 times more potent than kojic acid (48.62 ± 3.38 μM). Compounds 1c (IC_50_ = 18.09 ± 0.25 μM) and 2a (IC_50_ = 15.69 ± 0.40 μM) also suppressed tyrosinase activity more than kojic acid.

### 2.9. Kinetic Analyses of the Urolithin and Reduced Urolithin Derivatives

To examine the inhibitory modes of 1c, 1h, and 2a, we performed a kinetic study on mushroom tyrosinase using L-tyrosine as substrate. The mechanism of tyrosinase inhibition was determined by Lineweaver–Burk analysis (Figure 10 and Table 4). The inhibitory kinetics studies showed ki values of 1.9 × 10^−6^, 1.9 × 10^−6^, or 2.0 × 10^−6^ M, and 3.4 × 10^−6^, 3.1 × 10^−6^, or 2.6 × 10^−6^ M at 10, 20, or 40 μM for 1c and 2a, respectively, and of 4.1 × 10^−7^, 3.0 × 10^−7^, or 3.0 × 10^−7^ M at 2.5, 5, or 10 μM for 1h, respectively. K_M_ values were 12.54, 22.24, or 41.56 mM and 7.83, 15.07, or 32.42 mM at 10, 20, or 40 μM of 1c and 2a, respectively, and 5.64, 13.92, or 26.97 mM at 2.5, 5, or 10 μM for 1h, respectively. V_max_ values were 2.7 × 10^−2^, 3.1 × 10^−2^, and 2.9 × 10^−2^ mM·min^−1^ for 1c, 1h, and 2a, respectively, and these values were not dependent on inhibitor concentration. Lineweaver–Burk plots of the three compounds had similar patterns. Lines generated at each inhibitor concentration merged at one point on the *y*-axis. As inhibitor concentration increased, tyrosinase K_M_ value also increased gradually in a concentration-dependent manner without changing V_max_. These results indicate that 1c, 1h, and 2a are all competitive inhibitors of tyrosinase and utilize the same binding pocket as the substrate for tyrosinase.

### 2.10. Cell Viabilities of 1c and 1h and of 2a in B16F10 Melanoma Cells

EZ-cytox assays were performed to examine whether these compounds are cytotoxic. Murine B16F10 melanoma cells were treated with six different concentrations (0, 1, 2, 5, 10, or 20 µM) of the test compounds for 24 h in a humidified atmosphere, and optical densities were measured using a microplate reader.

The effects of 1c, 1h, and 2a on cell viabilities are summarized in Figure 11. No significant cytotoxic effect was found in melanoma cells at concentrations up to 20 µM for 2a. However, 1c and 1h were cytotoxic in the range 1–20 µM; at 20 µM, respective cell viabilities were 71.38% and 70.78% (According to the report by Su and co-workers, urolithin A and urolithin B exhibited higher cytotoxicity at the same concentration). This result indicates only 2a was non-toxic in murine melanoma B16F10 cells. Therefore, cell-based assays of melanin contents and cellular tyrosinase activities were carried out using only the reduced urolithin analog 2a at concentrations below 20 µM.

### 2.11. Tyrosinase Inhibitory Activity of the Reduced Urolithin Analog 2a in α-MSH- Plus IBMX-Stimulated B16F10 Melanoma Cells

The ability of 2a to inhibit tyrosinase was evaluated using B16F10 melanoma cells stimulated with α-melanocyte-stimulating hormone (α-MSH) plus 3-isobutyl-1-methylxanthine (IBMX). Murine B16F10 melanoma cells were seeded in 6-well culture plates and treated with 2a at 0, 5, 10, or 20 μM or with kojic acid (20 μM) for 3 h, and then α-MSH (1 μM) plus IBMX (200 μM) were added to enhance tyrosinase activities. After incubation for 48 h, tyrosinase-inhibitory activities were measured by measuring the optical densities of 2a spectrophotometrically.

As depicted in Figure 12, cellular tyrosinase activity was enhanced by α-MSH plus IBMX treatment and was decreased significantly and concentration-dependently by 2a. The inhibitory effect of 2a at 20 μM was much higher than that of kojic acid at 20 µM, and was greater at 10 μM than kojic acid at 20 μM. At a concentration of 20 μM, 2a reduced tyrosinase activity to the untreated control levels. Because 2a was not cytotoxic below 20 μM (Figure 11), this inhibition of tyrosinase activity was attributed to direct tyrosinase inhibition rather than to the cytotoxic effect of 2a. The docking simulation results shown in Figure 4, Figure 5 and Figure 6 also supported the notion that 2a directly inhibited tyrosinase by binding to its active site.

### 2.12. Inhibition of Melanin Production by the Reduced Urolithin Analog **2a** in α-MSH Plus IBMX Stimulated B16F10 Melanoma Cells

To explore the inhibitory effect of 2a on melanin production in B16F10 melanoma cells, cells were seeded in 6-well culture plates and treated with 2a at 0, 5, 10, or 20 μM, or kojic acid (20 μM) for 3 h. α-MSH and IBMX were then added and incubated for 48 h. Melanin contents in B16F10 cells were assessed by measuring optical densities spectrophotometrically.

The inhibitory effect of 2a on melanin production is shown in Figure 13. Compound 2a reduced melanin contents in α-MSH plus IBMX stimulated B16F10 melanoma cells more so than kojic acid at 20 µM. α-MSH plus IBMX treatment increased melanin content 3.58-fold and 2a reduced melanin levels significantly and concentration-dependently. Interestingly compound 2a at 5 µM inhibited melanin contents more potently than kojic acid at 20 µM. Considering the abilities of 2a to inhibit tyrosinase activity (Figure 12) and bind to the tyrosinase active site (Figure 4, Figure 5 and Figure 6), the observed inhibitory effect of 2a on melanogenesis appears to be mainly due to the inhibition of tyrosinase activity. However, even considering the degree of tyrosinase inhibitory activity of compound 2a, 2a showed a decrease in melanin contents to a much greater extent than kojic acid. Studies such as cell signaling inhibition associated with tyrosinase expression and cell permeability for mechanisms of action in which 2a exhibits superior melanin inhibitory activity compared to tyrosinase inhibitory activity will be published in due course.

## 3. Conclusions

We previously reported that the (*E*)-β-phenyl-α,β-unsaturated carbonyl motif plays a critical role in tyrosinase inhibition and the presence of 4-substituted resorcinol on the β-phenyl of this scaffold substantially increases tyrosinase inhibitory activity. To investigate whether the 4-substituted resorcinol entity in the absence of the α,β-unsaturated carbonyl scaffold inhibits tyrosinase, ten urolithin derivatives 1a–1j were prepared using the Hurtley reaction. Two urolithin derivatives containing 4-substituted resorcinol, that is, 1c and 1h, inhibited tyrosinase more strongly than kojic acid. However, replacing the two hydroxyl groups of 1c and 1h by two methoxyl groups resulted in complete loss of inhibitory activity. Substitution at the 8-position of 1c sensitively affected tyrosinase inhibition. The introduction of an 8-hydroxyl group in 1c significantly decreased tyrosinase inhibitory activity from 69% to 13%, whereas the introduction of a methoxyl group increased tyrosinase inhibition to 95%. Reduction of the lactone ring of the urolithin derivatives provided nine *o*-phenylbenzyl alcohol derivatives 2a–2i. Compound 2a containing 4-substituted resorcinol most potently inhibited tyrosinase, which was greater than that achieved by kojic acid. The conversion of 4-substituted resorcinol to the 2-substituted phloroglucinol greatly reduced tyrosinase-inhibitory activity (2a vs. 2c). These results indicate that the 4-substituted resorcinol motif importantly confers tyrosinase-inhibiting activity. Docking simulation and pharmacophore analysis of 1c, 1h, and 2a, which all contained the 4-substituted resorcinol, with mushroom and human tyrosinases revealed that two hydroxyl substituents of the 4-substituted resorcinol formed hydrogen bonds with the amino acid residues existing in tyrosinase active site. Kinetic studies indicated that 1c, 1h, and 2a are all competitive inhibitors. Since EZ-cytox assays showed that 1c and 1h were weakly cytotoxic to murine B16F10 melanoma cells, only 2a was used in cell-based assay experiments. Compound 2a at concentrations ≤ 20 µM was found to inhibit cellular tyrosinase activity and melanogenesis effectively and dose-dependently in α-MSH plus IBMX stimulated murine B16F10 melanoma cells. These results show 2a with a 4-substituted resorcinol is a promising novel candidate for the treatment of skin pigment disorders, and that tyrosinase inhibition can be significantly increased with the 4-substituted resorcinol entity alone without the α,β-unsaturated carbonyl motif.

## 4. Materials and Methods

### 4.1. General Methods

All chemicals were obtained commercially (Sejinci Co., Seoul, Korea, Aldrich Co., St. Louis, MO, USA and Alfa Aesar Co., Ward Hill, MA, USA) and used without further purification. Anhydrous solvents were distilled over CaH_2_ or Na/benzophenone before use. Reactions were monitored by thin-layer chromatography (TLC) on glass plates coated with silica gel using a fluorescent indicator (TLC Silica Gel 60 F254, Merck, Kenilworth, NJ, USA) and column chromatography was conducted on MP Silica 40–63, 60 Å. Low-resolution mass spectroscopy was performed in ESI negative mode using an Expression CMS spectrometer (Advion, Ithaca, NY, USA). High resolution mass spectroscopy data were obtained using a combined liquid chromatograph Agilent Accurate Mass Q-TOF (quadruple-time of flight) mass spectrometer (Agilent, Santa Clara, CA, USA) in ESI negative or positive mode. Nuclear magnetic resonance (NMR) spectra were obtained using a Varian Unity AS500 spectrometer and a Varian Unity INOVA 400 spectrometer (Agilent Technologies, Santa Clara, CA, USA) at 500 MHz and 400 MHz ^1^H-NMR, respectively, and on a Varian Unity INOVA 400 spectrometer at 100 MHz ^13^C-NMR. CDCl_3_ (*δ*_C_ 77.0 ppm and *δ*_H_ 7.24 ppm), and DMSO-*d*_6_ (*δ*_C_ 39.7 ppm and *δ*_H_ 2.50 ppm) were used as solvents for NMR samples. Chemical shift (*δ*) and coupling constant (*J*) values were measured in parts per million (ppm) and hertz (Hz), respectively. The abbreviations used for ^1^H NMR data were as follows; s (singlet), d (doublet), t (triplet), dd (doublet of doublets), m (multiplet), and brs (broad singlet).

#### 4.1.1. General Procedure Used for Synthesizing Urolithin Derivatives 1a, 1c, and 1g–1h

A solution of 2-bromobenzoic acid or 2-bromo-5-methoxybenzoic acid (100 mg), phloroglucinol or resorcinol (1.0 equiv.), and NaOH (2.0 equiv.) in water (1.0 mL) was heated at 70 °C for 10 min and then 5% CuSO_4_ aqueous solution (2.0 mL/1.0 g of benzoic acid) was added. The reaction mixture was refluxed for 40 min–9 h, cooled, acidified to pH 2 using 2M-HCl, and filtered, and the solid obtained was purified by silica gel column chromatography using methylene chloride and methanol (20:1–40:1) as eluant to give compounds **1a**, **1c**, and **1g**–1**h** as solids in yields of 16–36%.

*3-Hydroxy-6H-benzo[c]chromen-6-one (**1a**, urolithin B)*. Beige powder; reaction time, 40 min; 16% yield; ^1^H NMR (500 MHz, DMSO-*d*_6_) *δ* 10.31 (brs, 1H, OH), 8.23 (d, 1H, *J* = 8.0 Hz, 10-H), 8.16 (d, 1H, *J* = 8.5 Hz, 7-H), 8.14 (t, 1H, *J* = 8.5 Hz, 1-H), 7.86 (t, 1H, *J* = 8.0 Hz, 9-H), 7.53 (d, 1H, *J* = 8.0 Hz, 8-H), 6.82 (dd, 1H, *J* = 8.5, 2.5 Hz, 2-H), 6.72 (d, 1H, *J* = 2.5 Hz, 4-H); ^13^C NMR (100 MHz, DMSO-*d*_6_) *δ* 161.2, 160.6, 152.8, 135.9, 135.8, 130.3, 128.3, 125.5, 122.3, 119.6, 113.8, 110.0, 103.6; LRMS (ESI-) *m*/*z* 211 (M-H)^−^, 167 (M-H-CO_2_)^−^.

*1,3-Dihydroxy-6H-benzo[c]chromen-6-one (**1c**)*. Light ocher solid; reaction time, 9 h; yield, 31%; ^1^H NMR (500 MHz, DMSO-*d*_6_) *δ* 10.87 (s, 1H, OH), 10.12 (s, 1H, OH), 8.94 (d, 1H, *J* = 8.5 Hz, 10-H), 8.16 (d, 1H, *J* = 8.5 Hz, 7-H), 7.80 (t, 1H, *J* = 8.5 Hz, 9-H), 7.47 (t, 1H, *J* = 8.5 Hz, 8-H), 6.39 (d, 1H, *J* = 2.0 Hz, 4-H), 6.25 (d, 1H, *J* = 2.0 Hz, 2-H); ^13^C NMR (100 MHz, DMSO-*d*_6_) *δ* 161.4, 159.9, 158.3, 154.0, 136.1, 135.6, 130.0, 127.0, 126.5, 119.1, 100.5, 99.3, 95.8; LRMS (ESI-) *m*/*z* 227 (M-H)^−^, 183 (M-H-CO_2_)^−^.

*3-Hydroxy-8-methoxy-6H-benzo[c]chromen-6-one (**1g**)*. Ocher powder; reaction time, 6 h; 27% yield; ^1^H NMR (500 MHz, DMSO-*d*_6_) *δ* 10.17 (s, 1H, OH), 8.16 (d, 1H, *J* = 8.5 Hz, 10-H), 8.04 (d, 1H, *J* = 8.0 Hz, 1-H), 7.56 (d, 1H, *J* = 2.5 Hz, 7-H), 7.45 (dd, 1H, *J* = 8.5, 2.5 Hz, 9-H), 6.79 (dd, 1H, *J* = 8.0, 2.0 Hz, 2-H), 6.71 (d, 1H, *J* = 2.0 Hz, 4-H), 3.86 (s, 3H, OCH_3_); ^13^C NMR (100 MHz, DMSO-*d*_6_) *δ* 161.1, 159.6, 159.2, 151.8, 129.2, 124.8, 124.6, 124.2, 120.7, 113.8, 111.6, 110.2, 103.5, 56.2; LRMS (ESI-) *m*/*z* 226 (M-H-CH_3_)^−^.

*1,3-Dihydroxy-8-methoxy-6H-benzo[c]chromen-6-one (**1h**)*. Beige powder; reaction time, 9 h; 36% yield; ^1^H NMR (500 MHz, DMSO-*d*_6_) *δ* 10.75 (s, 1H, OH), 9.99 (s, 1H, OH), 8.88 (d, 1H, *J* = 9.0 Hz, 10-H), 7.60 (d, 1H, *J* = 3.0 Hz, 7-H), 7.42 (dd, 1H, *J* = 9.0, 3.0 Hz, 9-H), 6.37 (d, 1H, *J* = 2.0 Hz, 4-H), 6.24 (d, 1H, *J* = 2.0 Hz, 2-H), 3.85 (s, 3H, OCH_3_); ^13^C NMR (100 MHz, DMSO-*d*_6_) *δ* 161.3, 159.0, 158.0, 157.4, 153.1, 129.5, 128.4, 123.8, 120.3, 111.6, 100.5, 99.4, 95.6, 56.1; LRMS (ESI-) *m*/*z* 257 (M-H)^−^, 242 (M-H-CH_3_)^−^.

#### 4.1.2. General Procedure Used for Synthesizing Urolithin Derivatives 1b, 1d, 1i, and 1j

To a stirred solution containing an urolithin derivative 1a, 1c, or 1g–1h (50 mg) and K_2_CO_3_ (3.0–6.0 equiv.) in acetone (3.0 mL) was added iodomethane (1.2–4.8 equiv.) at 0 °C. The reaction mixture was stirred at ambient temperature for 11–73 h and partitioned between methylene chloride and water. The organic layer so obtained was dried over anhydrous MgSO_4_, filtered, and evaporated in vacuo to give pure urolithin derivatives 1b, 1d, 1i, and 1j as solids in yields of 35–94%.

*3-Methoxy-6H-benzo[c]chromen-6-one (**1b**)*. Yellowish solid; reaction time, 11 h; yield, 94%; ^1^H NMR (500 MHz, DMSO-*d*_6_) *δ* 8.30 (d, 1H, *J* = 8.0 Hz, 10-H), 8.24 (d, 1H, *J* = 8.5 Hz, 1-H), 8.18 (d, 1H, *J* = 8.0 Hz, 7-H), 7.89 (t, 1H, *J* = 8.0 Hz, 9-H), 7.58 (t, 1H, *J* = 8.0 Hz, 8-H), 6.99 (d, 1H, *J* = 2.5 Hz, 4-H), 6.98 (dd, 1H, *J* = 8.5, 2.5 Hz, 2-H), 3.84 (s, 3H, OCH_3_); ^13^C NMR (100 MHz, DMSO-*d*_6_) *δ* 161.9, 161.1, 152.8, 136.0, 135.4, 130.3, 128.7, 125.4, 122.6, 119.9, 113.0, 111.3, 102.2, 56.5; HRMS (ESI+) *m*/*z* C_14_H_11_O_3_ (M+H)^+^ calcd 227.0703, obsd 227.0702.

*1,3-Dimethoxy-6H-benzo[c]chromen-6-one (**1d**)*. Light yellowish powder; reaction time, 11 h; 93% yield; ^1^H NMR (500 MHz, DMSO-*d*_6_) *δ* 8.84 (d, 1H, *J* = 8.0 Hz, 10-H), 8.21 (d, 1H, *J* = 8.0 Hz, 7-H), 7.85 (t, 1H, *J* = 8.0 Hz, 9-H), 7.55 (t, 1H, *J* = 8.0 Hz, 8-H), 6.64 (s, 1H, *J* = 1.5 Hz), 6.61 (d, 1H, *J* = 1.5 Hz), 4.00 (s, 3H, OCH_3_), 3.85 (s, 3H, OCH_3_); ^13^C NMR (100 MHz, DMSO-*d*_6_) *δ* 161.7, 161.0, 159.8, 153.8, 135.8, 135.0, 130.2, 127.9, 126.7, 119.7, 96.7, 94.9, 57.0, 56.4; LRMS (ESI-) *m*/*z* 241 (M-CH_3_)^−^, 226 (M-2CH_3_)^−^.

*3,8-Dimethoxy-6H-benzo[c]chromen-6-one (**1i**)*. Yellowish powder; reaction time, 46 h; 35% yield; ^1^H NMR (500 MHz, CDCl_3_) *δ* 7.92 (d, 1H, *J* = 9.0 Hz, 10-H), 7.86 (d, 1H, *J* = 9.0 Hz, 1-H), 7.77 (d, 1H, *J* = 3.0 Hz, 7-H), 7.37 (dd, 1H, *J* = 9.0, 3.0 Hz, 9-H), 6.90 (dd, 1H, *J* = 9.0, 2.5 Hz, 2-H), 6.86 (d, 1H, *J* = 2.5 Hz, 4-H), 3.92 (s, 3H, OCH_3_), 3.87 (s, 3H, OCH_3_); ^13^C NMR (100 MHz, CDCl_3_) *δ* 161.7, 160.9, 159.4, 151.9, 128.9, 124.6, 123.3, 123.0, 121.3, 112.5, 111.5, 111.3, 101.8, 55.9, 55.8; HRMS (ESI+) *m*/*z* C_15_H_13_O_4_ (M+H)^+^ calcd 257.0808, obsd 257.0808.

*1,3,8-Trimethoxy-6H-benzo[c]chromen-6-one (**1j**)*. Light ocher powder; reaction time, 73 h; 90% yield; ^1^H NMR (500 MHz, DMSO-*d*_6_) *δ* 8.74 (d, 1H, *J* = 9.5 Hz, 10-H), 7.62 (d, 1H, *J* = 3.0 Hz, 7-H), 7.41 (dd, 1H, *J* = 9.5, 3.0 Hz, 9-H), 6.59 (d, 1H, *J* = 2.0 Hz), 6.57 (d, 1H, *J* = 2.0 Hz), 3.97 (s, 3H, OCH_3_), 3.86 (s, 3H, OCH_3_), 3.82 (s, 3H, OCH_3_); ^13^C NMR (100 MHz, DMSO-*d*_6_) *δ* 160.9, 160.9, 159.0, 158.5, 152.8, 128.6, 128.3, 123.8, 121.0, 111.8, 101.6, 96.7, 94.8, 56.9, 56.4, 56.1; HRMS (ESI+) *m*/*z* C_16_H_15_O_5_ (M+H)^+^ calcd 287.0914, obsd 287.0914.

#### 4.1.3. General Synthetic Procedure for Urolithin Derivatives 1e and 1f

A suspension of urolithin derivative 1g or 1h (50 mg) and AlCl_3_ (5.0 equiv.) in chlorobenzene (2.0 mL) was refluxed for 5 h. After cooling, water was added to the reaction mixture and the resulting precipitate was filtered and washed with methylene chloride to give urolithin derivatives 1e and 1f as solids in yields of 73–79%.

*3,8-Dihydroxy-6H-benzo[c]chromen-6-one (**1e**, urolithin A)*. Brown solid; reaction time, 5 h; 79% yield; ^1^H NMR (400 MHz, DMSO-*d*_6_) *δ* 10.17 (s, 1H, OH), 10.10 (s, 1H, OH), 8.04 (d, 1H, *J* = 8.8 Hz, 10-H), 7.96 (d, 1H, *J* = 8.4 Hz, 1-H), 7.45 (d, 1H, *J* = 2.0 Hz, 7-H), 7.26 (dd, 1H, *J* = 8.8, 2.0 Hz, 9-H), 6.74 (dd, 1H, *J* = 8.4, 2.0 Hz, 2-H), 6.67 (d, 1H, *J* = 2.0 Hz, 4-H); ^13^C NMR (100 MHz, DMSO-*d*_6_) *δ* 161.2, 159.2, 157.6, 151.5, 127.6, 124.8, 124.4, 124.2, 120.8, 114.2, 113.7, 110.5, 103.5; LRMS (ESI-) *m*/*z* 227 (M-H)^−^.

*1,3,8-Trihydroxy-6H-benzo[c]chromen-6-one (**1f**)*. Brown solid; reaction time, 5 h; 73% yield; ^1^H NMR (400 MHz, DMSO-*d*_6_) *δ* 10.65 (s, 1H, OH), 10.02 (s, 1H, OH), 9.90 (s, 1H, OH), 8.75 (d, 1H, *J* = 8.8 Hz, 10-H), 7.47 (s, 1H, 7-H), 7.21 (d, 1H, *J* = 8.8 Hz, 9-H), 6.31 (s, 1H, 4-H), 6.18 (s, 1H, 2-H); ^13^C NMR (100 MHz, DMSO-*d*_6_) *δ* 166.1, 163.3, 161.9, 161.1, 157.6, 133.2, 132.6, 128.9, 125.2, 119.0, 105.1, 104.4, 100.3; LRMS (ESI-) *m*/*z* 243 (M-H)^−^.

#### 4.1.4. General Synthetic Procedure for *o*-phenylbenzyl Alcohol Derivatives 2a–2i

A solution of an urolithin derivative 1a–1e or 1g–1j (100 mg) in THF (8 mL) at ambient temperature was added dropwise to a stirred 1M THF solution of LiAlH_4_ (2.6–4.0 equiv.). After refluxing the reaction mixture for 3–24 h, it was cooled to 0 °C and water (1.0 mL/g on LiAlH_4_), 15 wt% NaOH (1.0 mL/g on LiAlH_4_), and water (3.0 mL/g on LiAlH_4_) were consecutively added. The precipitate generated was filtered off and the filtrate was acidified to pH 3 using 2M-HCl, partitioned between ethyl acetate and water, and the organic layer was dried over anhydrous MgSO_4_, filtered, and evaporated under reduced pressure. Compounds 2a–2b, 2e, and 2g were obtained as sticky oils or solids in yields of 47–76%. However, 2c–2d, 2f, 2h, and 2i were not pure and were purified by silica gel column chromatography using methylene chloride and methanol (20:1 or 10:1) as eluent to give pure urolithin derivatives 2c–2d, 2f, 2h, and 2i as sticky oils or solids in yields of 15–73%.

*2′-(Hydroxymethyl)-[1,1′-biphenyl]-2,4-diol (**2a**)*. Brown sticky oil; reaction time, 3 h; 49% yield; ^1^H NMR (500 MHz, DMSO-*d*_6_) *δ* 9.24 (s, 1H, OH), 9.17 (s, 1H, OH), 7.49 (d, 1H, *J* = 7.0 Hz, 6′-H), 7.26 (t, 1H, *J* = 7.0 Hz, 5′-H), 7.18 (t, 1H, *J* = 7.0 Hz, 4′-H), 7.02 (d, 1H, *J* = 7.0 Hz, 3′-H), 6.77 (d, 1H, *J* = 8.0 Hz, 6-H), 6.35 (d, 1H, *J* = 2.0 Hz, 3-H), 6.24 (dd, 1H, *J* = 8.0, 2.0 Hz, 5-H), 4.89 (brs, 1H, CH_2_O*H*), 4.31 (s, 2H, CH_2_); ^13^C NMR (100 MHz, DMSO-*d*_6_) *δ* 158.3, 155.7, 141.6, 137.6, 131.8, 130.7, 126.9, 126.7, 126.5, 119.2, 107.0, 103.1, 61.3; LRMS (ESI-) *m*/*z* 215 (M-H)^−^, 213 (M-H-H_2_)^−^; HRMS (ESI-) *m*/*z* C_13_H_11_O_3_ (M-H)^−^ calcd 215.0714, obsd 215.0716.

*2′-(Hydroxymethyl)-4-methoxy-[1,1′-biphenyl]-2-ol (**2b**)*. Beige solid; reaction time, 5 h; 67% yield; ^1^H NMR (500 MHz, DMSO-*d*_6_) *δ* 9.38 (s, 1H, OH), 7.50 (d, 1H, *J* = 7.5 Hz, 6′-H), 7.28 (t, 1H, *J* = 7.5 Hz, 5′-H), 7.19 (t, 1H, *J* = 7.5 Hz, 4′-H), 7.03 (d, 1H, *J* = 7.5 Hz, 3′-H), 6.91 (d, 1H, *J* = 8.5 Hz, 6-H), 6.45 (d, 1H, *J* = 2.0 Hz, 3-H), 6.42 (dd, 1H, *J* = 8.5, 2.0 Hz, 5-H), 4.90 (brs, 1H, CH_2_O*H*), 4.30 (brs, 2H, CH_2_), 3.71 (s, 3H, OCH_3_); ^13^C NMR (100 MHz, DMSO-*d*_6_) *δ* 160.3, 155.9, 141.5, 137.2, 131.9, 130.6, 127.2, 126.8, 126.6, 120.9, 105.1, 101.9, 61.3, 55.6; LRMS (ESI-) *m*/*z* 229 (M-H)^−^, 227 (M-H-H_2_)^−^.

*2′-(Hydroxymethyl)-[1,1′-biphenyl]-2,4,6-triol (**2c**)*. Light brown sticky oil; reaction time, 8 h; 31% yield; ^1^H NMR (500 MHz, DMSO-*d*_6_) *δ* 8.99 (s, 1H, OH), 8.79 (s, 2H, OH), 7.45 (d, 1H, *J* = 8.0 Hz, 6′-H), 7.20 (t, 1H, *J* = 8.0 Hz, 5′-H), 7.11 (t, 1H, *J* = 8.0 Hz, 4′-H), 6.95 (d, 1H, *J* = 8.0 Hz, 3′-H), 5.84 (s, 2H, 3-H, 5-H), 4.79 (t, 1H, *J* = 5.5 Hz, CH_2_O*H*), 4.26 (d, 2H, *J* = 5.5 Hz, CH_2_); ^13^C NMR (100 MHz, DMSO-*d*_6_) *δ* 158.0, 156.5, 142.4, 133.8, 131.9, 126.5, 126.0, 125.9, 106.8, 94.9, 61.4; LRMS (ESI-) *m*/*z* 231 (M-H)^−^.

*2′-(Hydroxymethyl)-4,6-dimethoxy-[1,1′-biphenyl]-2-ol (**2d**)*. Yellowish solid; reaction time, 6 h; 73% yield; ^1^H NMR (500 MHz, DMSO-*d*_6_) *δ* 9.18 (brs, 1H, OH), 7.47 (d, 1H, *J* = 7.0 Hz, 6′-H), 7.23 (t, 1H, *J* = 7.0 Hz, 5′-H), 7.14 (t, 1H, *J* = 7.0 Hz, 4′-H), 6.93 (d, 1H, *J* = 7.0 Hz, 3′-H), 6.11 (d, 1H, *J* = 2.0 Hz), 6.10 (d, 1H, *J* = 2.0 Hz), 4.84 (brs, 1H, CH_2_O*H*), 4.23 (d, 1H, *J* = 14.0 Hz, C*H*H), 4.23 (d, 1H, *J* = 14.0 Hz, CH*H*), 3.71 (s, 3H, OCH_3_), 3.56 (s, 3H, OCH_3_); ^13^C NMR (100 MHz, DMSO-*d*_6_) *δ* 160.6, 158.8, 156.4, 142.3, 132.9, 131.6, 126.9, 126.1, 125.9, 109.1, 94.4, 90.5, 61.2, 56.0, 55.6; LRMS (ESI-) *m*/*z* 259 (M-H)^−^, 257 (M-H-H_2_)^−^.

*2′-(Hydroxymethyl)-[1,1′-biphenyl]-2,4,4′-triol (**2e**)*. Brown sticky oil; reaction time, 14 h; 76% yield; ^1^H NMR (500 MHz, DMSO-*d*_6_) *δ* 9.14 (s, 1H, OH), 9.13 (s, 1H, OH), 9.01 (s, 1H, OH), 6.92 (d, 1H, *J* = 2.0 Hz, 3′-H), 6.79 (d, 1H, *J* = 8.0 Hz, 6′-H), 6.70 (d, 1H, *J* = 8.0 Hz, 6-H), 6.55 (dd, 1H, *J* = 8.0, 2.0 Hz, 5′-H), 6.30 (d, 1H, *J* = 1.5 Hz, 3-H), 6.19 (dd, 1H, *J* = 8.0, 1.5 Hz, 5-H), 4.82 (t, 1H, *J* = 5.5 Hz, CH_2_O*H*), 4.21 (brs, 2H, CH_2_); ^13^C NMR (100 MHz, DMSO-*d*_6_) *δ* 157.9, 156.7, 155.9, 142.8, 132.0, 131.6, 128.1, 119.2, 113.5, 113.4, 106.8, 103.0, 61.4; LRMS (ESI-) *m*/*z* 231 (M-H)^−^, 229 (M-H-H_2_)^−^, 227 (M-H-2H_2_)^−^.

*2′-(Hydroxymethyl)-4′-methoxy-[1,1′-biphenyl]-2,4-diol (**2f**)*. Light brown sticky oil; reaction time, 24 h; 15% yield; ^1^H NMR (500 MHz, DMSO-*d*_6_) *δ* 9.18 (brs, 1H, OH), 9.09 (s, 1H, OH), 7.05 (d, 1H, *J* = 2.0 Hz, 3′-H), 6.93 (d, 1H, *J* = 8.5 Hz, 6′-H), 6.74 (dd, 1H, *J* = 8.5, 2.0 Hz, 5′-H), 6.73 (d, 1H, *J* = 8.5 Hz, 6-H), 6.32 (d, 1H, *J* = 2.0 Hz, 3-H), 6.21 (dd, 1H, *J* = 8.5, 2.0 Hz, 5-H), 4.92 (brs, 1H, CH_2_O*H*), 4.26 (brs, 2H, CH_2_), 3.74 (s, 3H, OCH_3_); ^13^C NMR (100 MHz, DMSO-*d*_6_) *δ* 158.8, 158.1, 155.9, 143.0, 132.0, 131.7, 129.8, 122.2, 118.8, 112.0, 106.9, 103.1, 61.4, 55.6; LRMS (ESI-) *m*/*z* 245 (M-H)^−^, 243 (M-H-H_2_)^−^.

*2′-(Hydroxymethyl)-4,4′-dimethoxy-[1,1′-biphenyl]-2-ol (**2g**)*. Brown sticky oil; reaction time, 10 h; 47% yield; ^1^H NMR (500 MHz, DMSO-*d*_6_) *δ* 9.31 (s, 1H, OH), 7.06 (d, 1H, *J* = 2.0 Hz, 3′-H), 6.95 (d, 1H, *J* = 8.0 Hz, 6′-H), 6.87 (d, 1H, *J* = 8.5 Hz, 6-H), 6.76 (dd, 1H, *J* = 8.0, 2.0 Hz, 5′-H), 6.43 (d, 1H, *J* = 2.0 Hz, 3-H), 6.39 (dd, 1H, *J* = 8.5, 2.0 Hz, 5-H), 4.94 (t, 1H, *J* = 5.5 Hz, CH_2_O*H*), 4.26 (brs, 2H, CH_2_), 3.75 (s, 3H, OCH_3_), 3.70 (s, 3H OCH_3_); ^13^C NMR (100 MHz, DMSO-*d*_6_) *δ* 160.1, 158.9, 156.0, 143.0, 132.1, 131.6, 129.4, 120.5, 112.0, 105.0, 104.7, 101.9, 61.3, 55.6, 55.6; LRMS (ESI-) *m*/*z* 259 (M-H)^−^, 257 (M-H-H_2_)^−^.

*2′-(Hydroxymethyl)-4′-methoxy-[1,1′-biphenyl]-2,4,6-triol (**2h**)*. Brown sticky oil; reaction time, 8 h; 41% yield; ^1^H NMR (500 MHz, DMSO-*d*_6_) *δ* 8.94 (s, 1H, OH), 8.70 (s, 2H, OH), 7.03 (d, 1H, *J* = 2.0 Hz, 3′-H), 6.85 (d, 1H, *J* = 8.0 Hz, 6′-H), 6.69 (dd, 1H, *J* = 8.0, 2.0 Hz, 5′-H), 5.83 (s, 2H, 3-H, 5-H), 4.83 (t, 1H, *J* = 5.5 Hz, CH_2_O*H*), 4.22 (d, 2H, *J* = 5.5 Hz, CH_2_), 3.73 (s, 3H, OCH_3_); ^13^C NMR (100 MHz, DMSO-*d*_6_) *δ* 158.6, 157.9, 156.7, 143.8, 132.8, 125.7, 111.6, 111.3, 106.3, 94.9, 61.4, 55.5; LRMS (ESI-) *m*/*z* 261 (M-H)^−^.

*2′-(Hydroxymethyl)-4,4′,6-trimethoxy-[1,1′-biphenyl]-2-ol (**2i**)*. Yellowish solid; reaction time, 6 h; 63% yield; ^1^H NMR (500 MHz, DMSO-*d*_6_) *δ* 9.09 (s, 1H, OH), 7.05 (d, 1H, *J* = 2.0 Hz, 3′-H), 6.84 (d, 1H, *J* = 8.0 Hz, 6′-H), 6.71 (dd, 1H, *J* = 8.0, 2.0 Hz, 5′-H), 6.10 (s, 1H, 5-H), 6.08 (s, 1H, 3-H), 4.89 (t, 1H, *J* = 5.5 Hz, CH_2_O*H*), 4.18 (dd, 1H, *J* = 14.5, 5.5 Hz, C*H*H), 4.12 (dd, 1H, *J* = 14.5, 5.5 Hz, CH*H*), 3.74 (s, 3H, OCH_3_), 3.70 (s, 3H, OCH_3_), 3.55 (s, 3H, OCH_3_); ^13^C NMR (100 MHz, DMSO-*d*_6_) *δ* 160.5, 159.0, 158.8, 156.6, 143.7, 132.6, 124.8, 111.7, 111.4, 108.8, 94.4, 90.5, 61.2, 55.9, 55.6, 55.5; LRMS (ESI-) *m*/*z* 289 (M-H)^−^, 287 (M-H-H_2_)^−^.

### 4.2. Biological Studies

#### 4.2.1. Mushroom Tyrosinase Assay

Mushroom tyrosinase inhibitory assays were performed on compounds 1a–1j and 2a–2i as previously described with slight modification [73]. Briefly, a 200 µL mixture [170 µL of substrate solution (potassium phosphate buffer 14.7 mM + 293 µM L-tyrosine solution), 20 µL of tyrosinase, and 10 µL of 1a–1j or 2a–2i (final concentration: 50 μM) or kojic acid (50 μM)] was added to a 96-well microplate and incubated at 37 °C for 30 min. Absorbances were measured at 475 nm using a microplate reader (VersaMaxTM, Molecular Devices, Sunnyvale, CA, USA). Kojic acid was used as the positive control. The experiment was repeated three times. Mushroom tyrosinase inhibitions % were calculated using:% Inhibition = [1 − (A/B)] × 100(1)
where A is the absorbance of the test compound and B is the absorbance of the blank.

To determine the IC_50_ values of tyrosinase inhibition by 1c, 1h, 2a, or kojic acid, dose-dependent inhibition experiments were performed in triplicate at 3–5 different concentrations. Log-linear curves were plotted, and their equations were determined. IC_50_ was defined as the concentration that inhibited tyrosinase activity by 50%.

#### 4.2.2. Kinetic Analysis Studies: Lineweaver–Burk Plots

Kinetic studies on compounds 1c, 1h and 2a were performed as previously described with slight modification [74]. Briefly, 170 μL of a mixture of an aqueous solution of L-tyrosine 1.0, 2.0, 4.0, or 8.0 mM and 50 mM of potassium phosphate (pH 6.5) was added to a 96-well plate. To this was added 20 μL of aqueous mushroom tyrosinase solution (200 units) and 10 μL of test compounds (final concentrations: 10, 20, and 40 μM for 1c and 2a, and 2.5, 5, and 10 μM for 1h). Dopachrome levels were monitored during the reaction using a microplate reader by measuring increases in absorbance at 475 nm. Lineweaver–Burk plots were used to determine V_max_ (maximal reaction rates) and K_M_ (Michaelis–Menten constants) of mushroom tyrosinase activities. The plots of the inverse of reaction rate (1/V) versus the inverse of substrate concentrations (1/[S]) were drawn and used to determined mechanisms based on positions of line convergences.

#### 4.2.3. Docking and Molecular Modeling Studies of 1c, 1h, 2a, and Kojic acid in Mushroom Tyrosinase

Docking and molecular modeling studies were performed between tyrosinase and 1c, 1h, 2a, or kojic acid, as previously described with some modification [75,76]. In short, the 3D structures of 1c, 1h, and 2a or kojic acid were created in Chem3D Pro 12.0 software. Chimera, AutoDock Vina 1.1.2, AutoDock 4 [75], and Dock 6 [77] were used to calculate docking scores between 1c, 1h, 2a, or kojic acid and tyrosinase. The 3D structure of tyrosinase (*Agaricus bisporus*) was obtained from the Protein Data Bank (ID: 2Y9X). LigandScout 4.2.1 was used to create pharmacophore models representing possible interactions between ligands and amino acid residues of tyrosinase.

#### 4.2.4. Docking and Molecular Studies of 1c, 1h, 2a, and Kojic Acid Using Human Tyrosinase Homology Model

The human tyrosinase homology model was created using the Swiss-Model online server and Schrödinger Suite. The full sequence of human tyrosinase P14679 was obtained from UniProt, a freely accessible protein sequence database. The protein sequences of P14679 were processed in Swiss-Model using various protein templates. The crystal structure of human tyrosinase-related protein 1 (hTYRP1, PDB: 5M8Q) was identified as the most identical protein template. The homology model was built using protein template of 5M8Q. The newly created model was downloaded from Swiss-Model in PDB format and further prepared for ligand docking using the Schrodinger Suite. The structures of ligands 1c, 1h, 2a, and kojic acid were imported in CDXML format into the Schrödinger Maestro entry list, prepared with LigPrep [78], and docked against human tyrosinase model. Docking scores were obtained using glide and 2D and 3D structures of all ligand–protein complexes were generated [79].

#### 4.2.5. B16F10 Melanoma Cell Culture

B16F10 murine melanoma cells were purchased from the American Type Culture Collection (ATCC, Manassas, VA, USA). Cells were cultured in Dulbecco’s modified Eagle’s medium (10% fetal bovine serum (FBS), and 1% streptomycin) obtained from Gibco/Thermo Fisher Scientific (Carlsbad, CA, USA) at 37 °C in a 5% CO_2_ environment. These cultured cells were used for cell viability, melanin content, and cellular tyrosinase inhibitory assays.

#### 4.2.6. Cell Cytotoxicity Studies: EZ-Cytox Assay

EZ-cytox assays were performed on 1c, 1h, and 2a, according to previously described [80]. B16F10 cells were seeded in a 96-well plate at a density of 5 × 10^4^ cells/well and incubated at 37 °C in a humidified 5% CO_2_ atmosphere overnight. Cells were then treated with different concentrations of 1c, 1h, or 2a (0, 1, 2, 5, 10, or 20 µM) and incubated for 24 h under the same conditions. EZ-cytox solution (10 µL) was then added and the plate was incubated for 3 h at 37 °C in a humidified 5% CO_2_ atmosphere. Optical densities of wells were measured at 450 nm using an ELISA reader. All experiments were independently repeated three times.

#### 4.2.7. Cellular Tyrosinase Inhibition in B16F10 Melanoma Cells

The tyrosinase inhibitory activity of 2a in B16F10 melanoma cells was analyzed as previously described with slight modification [81]. Briefly, B16F10 cells were pretreated with 2a at 0, 5, 10, or 20 μM or with kojic acid at 20 μM for 3 h and then treated with 1 μM of α-MSH plus 200 µM of IBMX for 48 h. Cells were washed with PBS two or three times, lysed with a 100 μL of buffer (90 μL of PBS (50 mM), 5 μL of PMSF (0.1 mM), and 5 μL of 1% Triton X-100), frozen at −80 °C for 30 min, and centrifuged at 12,000 rpm for 30 min at 4 °C. Lysate supernatants (80 µL) were mixed with 20 µL of L-dopa (2 mg/mL) in a 96-well plate and incubated at 37 °C for 15 min. Amounts of dopachrome in the cells were determined by measuring optical densities at 475 nm using a microplate reader (Tecan, Männedorf, Switzerland). All the experiments were independently repeated three times.

#### 4.2.8. Melanin Content Assays in B16F10 Melanoma Cells

The effect of compound 2a on melanin production was assessed using a melanin content assay as previously described with minor modification [82]. Briefly, B16F10 cells were pretreated with 0, 5, 10, or 20 μM of 2a or 20 μM of kojic acid for 3 h, 1 μM of α-MSH and 200 µM of IBMX were then added, and cells were incubated for 48 h. After washing cells with PBS two or three times, they were incubated for 1 h in 100 µL of 1N NaOH solution at 60 °C. Levels of melanin in cells were determined by measuring optical densities at 405 nm on a microplate reader. All experiments were independently repeated three times.

#### 4.2.9. Statistical Analysis

The statistical analysis was performed using Graph Pad Prism 5 software (La Jolla, CA, USA). Significances of differences were assessed using Tukey’s test and one-way ANOVA. Results are presented as means ± standard errors and statistical significance was accepted for *p* values < 0.05.

## Data Availability

Not applicable.

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
