# Peer review of "Urolithin and Reduced Urolithin Derivatives as Potent Inhibitors of Tyrosinase and Melanogenesis: Importance of the 4-Substituted Resorcinol Moiety"

_ijms, 2021, doi:10.3390/ijms22115616_

Round 1

Reviewer 1 Report

The manuscript provides substantial experimental data on the inhibition of mushroom tyrosinase with special focus on the 4-substituted resorcinol moiety.

The main problem with the manuscript is the lack of citing and discussing the current dermatological literature on tyrosinase inhibitors and the 4-substituted resorcinol derivatives currently being used in topical products for the treatment of hyperpigmentation.

Already the title implies that the manuscript is about human melanogenesis, however, neither human tyrosinase nor human melanocytes were used in the experiments.

The references about human hyperpigmentary disorders and approaches for treatment are outdated. From the 20 first references listed, 12 are more than 20 years old. This would not be a problem, if also the most recent literature would have been cited.

Several 4-substituted resorcinol derivatives are already in topical products for cosmetic or dermatological application: 4-butylresorcinol (Rucinol), 4-hexylresorcinol, phenylethyl resorcinol, dimethoxytolylpropyl resorcinol, isobutylamido-thiazolyl resorcinol (Thiamidol) to name only the most important.

The below references describe the above mentioned substances and the most recent advances in inhibition of human tyrosinase and, therefore, should be cited in a revision of the manuscript:

Advances in the Design of Genuine Human Tyrosinase Inhibitors for Targeting Melanogenesis and Related Pigmentations. Roulier B, Pérès B, Haudecoeur R.J Med Chem. 2020 Nov 25;63(22):13428-13443

Structure and Function of Human Tyrosinase and Tyrosinase-Related Proteins. Lai X, Wichers HJ, Soler-Lopez M, Dijkstra BW.Chemistry. 2018 Jan 2;24(1):47-55

Evaluation of efficacy and safety of rucinol serum in patients with melasma: a randomized controlled trial. Khemis A, Kaiafa A, Queille-Roussel C, Duteil L, Ortonne JP.Br J Dermatol. 2007 May;156(5):997-1004.

Phenylethyl resorcinol smartLipids for skin brightening - Increased loading & chemical stability. Köpke D, Müller RH, Pyo SM.Eur J Pharm Sci. 2019 Sep 1;137:104992.

Inhibition of Human Tyrosinase Requires Molecular Motifs Distinctively Different from Mushroom Tyrosinase. Mann T, Gerwat W, Batzer J, Eggers K, Scherner C, Wenck H, Stäb F, Hearing VJ, Röhm KH, Kolbe L.J Invest Dermatol. 2018 Jul;138(7):1601-1608.

Structure-Activity Relationships of Thiazolyl Resorcinols, Potent and Selective Inhibitors of Human Tyrosinase. Mann T, Scherner C, Röhm KH, Kolbe L.Int J Mol Sci. 2018 Feb 28;19(3):690.   

Before resubmission, the authors should test at least their most potent mushroom tyrosinase inhibitors on human melanocytes and/or tyrosinase preparations from human melanocytes and use some of the above mentioned inhibitors to compare the urolithins to the currrently used inhibitors.

Author Response

We deeply appreciate for the reviewer’s valuable comments. 

Point 1:

The manuscript provides substantial experimental data on the inhibition of mushroom tyrosinase with special focus on the 4-substituted resorcinol moiety.

The main problem with the manuscript is the lack of citing and discussing the current dermatological literature on tyrosinase inhibitors and the 4-substituted resorcinol derivatives currently being used in topical products for the treatment of hyperpigmentation.

Already the title implies that the manuscript is about human melanogenesis, however, neither human tyrosinase nor human melanocytes were used in the experiments.

The references about human hyperpigmentary disorders and approaches for treatment are outdated. From the 20 first references listed, 12 are more than 20 years old. This would not be a problem, if also the most recent literature would have been cited.

Several 4-substituted resorcinol derivatives are already in topical products for cosmetic or dermatological application: 4-butylresorcinol (Rucinol), 4-hexylresorcinol, phenylethyl resorcinol, dimethoxytolylpropyl resorcinol, isobutylamido-thiazolyl resorcinol (Thiamidol) to name only the most important.

The below references describe the above-mentioned substances and the most recent advances in inhibition of human tyrosinase and, therefore, should be cited in a revision of the manuscript:

Response: Thank you for valuable suggestions. We truly endorse your comments and thus we added the recommended references to our MS in the introduction section.

Advances in the Design of Genuine Human Tyrosinase Inhibitors for Targeting Melanogenesis and Related Pigmentations. Roulier B, Pérès B, Haudecoeur R.J Med Chem. 2020 Nov 25;63(22):13428-13443.

Response: Reference have been inserted as reference number 56 in the 3rd paragraph of the introduction.

Structure and Function of Human Tyrosinase and Tyrosinase-Related Proteins. Lai X, Wichers HJ, Soler-Lopez M, Dijkstra BW.Chemistry. 2018 Jan 2;24(1):47-55

Response: Reference have been inserted as reference number 57 in the 3rd paragraph of the introduction.

Evaluation of efficacy and safety of rucinol serum in patients with melasma: a randomized controlled trial. Khemis A, Kaiafa A, Queille-Roussel C, Duteil L, Ortonne JP.Br J Dermatol. 2007 May;156(5):997-1004.

Response: Reference have been inserted as reference number 15 in the 1st paragraph of the introduction.

Phenylethyl resorcinol smartLipids for skin brightening - Increased loading & chemical stability. Köpke D, Müller RH, Pyo SM.Eur J Pharm Sci. 2019 Sep 1;137:104992.

Response: Reference have been inserted as reference number 16 in the 1st paragraph of the introduction.

Inhibition of Human Tyrosinase Requires Molecular Motifs Distinctively Different from Mushroom Tyrosinase. Mann T, Gerwat W, Batzer J, Eggers K, Scherner C, Wenck H, Stäb F, Hearing VJ, Röhm KH, Kolbe L.J Invest Dermatol. 2018 Jul;138(7):1601-1608.

Response: Reference have been inserted as reference number 57 in the 3rd paragraph of the introduction.

Structure-Activity Relationships of Thiazolyl Resorcinols, Potent and Selective Inhibitors of Human Tyrosinase. Mann T, Scherner C, Röhm KH, Kolbe L.Int J Mol Sci. 2018 Feb 28;19(3):690. 

Response: Reference have been inserted as reference number 17 in the 1st paragraph of the introduction.

Point 2:

Before resubmission, the authors should test at least their most potent mushroom tyrosinase inhibitors on human melanocytes and/or tyrosinase preparations from human melanocytes and use some of the above-mentioned inhibitors to compare the urolithins to the currently used inhibitors.

Response 2: Thank you for your valuable comment. Our ultimate goal is to develop novel human tyrosinase inhibitors. But, the goal of this study is to search for promising candidates that are expected to have human tyrosinase inhibitory ability. Therefore, we first synthesized urolithin derivatives and reduced urolithin derivatives, and selected two compounds with a high probability of exhibiting mammalian tyrosinase inhibitory activity by investigating the mushroom tyrosinase inhibitory activity. In B16F10 melanoma cells, a mammalian tyrosinase inhibitory activity of these selected compounds was examined and also antimelanogenic effect in the cells was examined.

During the past decade, we found many resorcinol compounds showing mushroom tyrosinase-inhibitory activity and mammalian (B16F10 cellular) tyrosinase inhibitory activity more potently than kojic acid (see below Figure). As with the reviewer's comment, in the near future (Now, we have neither human tyrosinase nor human melanocytes. Please, kindly consider it.) we will assay the tyrosinase inhibitory activity of these compounds using human tyrosinase or human melanocytes and report the results along with a comparison of the currently used inhibitors.

Figure. Resorcinol compounds exhibiting both mushroom tyrosinase inhibition and mammalian tyrosinase inhibition more strongly than kojic acid (Please refer to the attached file)

References to Figure

Chung KW, Park YJ, Choi YJ, Park MH, Ha YM, Uehara Y, et al. Evaluation of in vitro and in vivo anti-melanogenic activity of a newly synthesized strong tyrosinase inhibitor (E)-3-(2, 4 dihydroxybenzylidene) pyrrolidine-2, 5-dione (3-DBP). Biochimica et Biophysica Acta (BBA)-General Subjects 2012;1820:962-69.

Kim CS, Noh SG, Park Y, Kang D, Chun P, Chung HY, et al. A potent tyrosinase inhibitor,(E)-3-(2, 4-Dihydroxyphenyl)-1-(thiophen-2-yl) prop-2-en-1-one, with anti-melanogenesis properties in α-MSH and IBMX-induced B16F10 melanoma cells. Molecules 2018;23:2725

Jung HJ, Noh SG, Ryu IY, Park C, Lee JY, Chun P, et al. (E)-1-(Furan-2-yl)-(substituted phenyl) prop-2-en-1-one Derivatives as Tyrosinase Inhibitors and Melanogenesis Inhibition: An In Vitro and In Silico Study. Molecules 2020;25:5460.

Ullah S, Park Y, Ikram M, Lee S, Park C, Kang D, et al. Design, synthesis and anti-melanogenic effect of cinnamamide derivatives. Bioorganic & medicinal chemistry 2018;26:5672-81.

Reviewer 2 Report

The presented reaults based on urolithin and reduced urolithin derivatives as potent inhibitors of tyrosinase and melanogenesis are quite interesting and promising. The paper shpuld be accepted for publication after minor revisions - for shure English should be corrected and polish. Additionally, I have to remarks:

  • why authors have used AutoDock if after that they have made the calculation with use of schrodinger?
  • could authors give the information on convergence of the location of the kojic acid after docking (for each method) compared to the crystal structure?
  • Could authors give the RMSD values?

Author Response

We deeply appreciate for the reviewer’s valuable comments.

Point 1:

The presented results based on urolithin and reduced urolithin derivatives as potent inhibitors of tyrosinase and melanogenesis are quite interesting and promising. The paper should be accepted for publication after minor revisions - for shure English should be corrected and polish.

Response 1: Thank you for your valuable encouraging comments. We have polished and corrected English mistakes in the MS and can be easily tracked.

Point 2:

Additionally, I have to remarks:

why have authors used AutoDock if after that they have made the calculation with use of Schrödinger?

Response 2: Thank you for your valuable opinion. The main purpose of using other software such as AutoDock Vina, AutoDock 4, and Dock 6 was to calibrate the docking results. Interestingly, we found that docking scores and protein-ligand interactions were consistent across all docking software.

Point 3:

could authors give the information on convergence of the location of the kojic acid after docking (for each method) compared to the crystal structure?

Response 3: Thank you very much for your comment. As shown in the following image (refer to the attached file), we compared kojic acid and the original ligand (tropolone) in the crystal structure of the mushroom tyrosinase enzyme 2Y9X. According to the superimpose structure of kojic acid and tropolone, it can be said that kojic acid (pink) occupies the same binding site as tropolone (white). Therefore, there is a strong convergence observed between kojic acid and the original ligand after docking.

Point 4:

Could authors give the RMSD values?

Response 4: Thank you for your valuable comment. We calculated the RMSD values of our compounds 1c, 1h, and 2a against kojic acid using Maestro. The values are compared as follow (refet to the attached file).

Round 2

Reviewer 1 Report

The authors adequately addressed the points raised in the first review.